



# Analysing surface energy balance closure and partitioning over a semi-arid savanna FLUXNET site in Skukuza, Kruger National Park, South Africa

Nobuhle P. Majozi[1,2], Chris M. Mannaerts[2], Abel Ramoelo[1,5], Renaud Mathieu[1,3], Alecia Nickless[4], Wouter Verhoef[2]

[1]Earth Observation Group, Natural Resources and Environment, Council for Scientific and Industrial Research, Pretoria, South Africa, 0001

[2]Department of Water Resources, Faculty of Geo-Information Science and Earth Observation (ITC), University of Twente, Enschede, 75AA, the Netherlands

[3]Department of Geography, Geoinformatics and Meteorology, University of Pretoria, South Africa

[4]Nuffield Department of Primary Care Health Sciences, University of Oxford, Oxford, OX2 6GG, United Kingdom

[5]University of Limpopo, Risk and Vulnerability Centre, Sovenga, South Africa, 0727

*Correspondence to*: N. P. Majozi (nmajozi@csir.co.za)

**Abstract**

Flux towers provide essential terrestrial climate, water and radiation budget information needed for environmental monitoring and evaluation of climate change impacts on ecosystems and society in general. They are also intended for calibration and validation of satellite-based earth observation and monitoring efforts, such as assessment of evapotranspiration from land and vegetation surfaces using surface energy balance approaches.

In this paper, 15 years of Skukuza eddy covariance data, i.e. from 2000 to 2014, were analysed for surface energy balance closure (EBC) and partitioning. The surface energy balance closure was evaluated using the ordinary least squares regression (OLS) of turbulent energy fluxes (sensible (H) and latent heat (LE)) against available energy (net radiation (Rn) less soil heat (G)), and the energy balance ratio (EBR). Partitioning of the surface energy during the wet and dry seasons was investigated, as well as how it is affected by atmospheric vapor pressure deficit (VPD), and net radiation.

After filtering years with bad data (2004-2008), our results show an overall mean EBR of 0.93. Seasonal variations of EBR also showed summer (0.98) and spring (1.02) were closest to unity, with winter (0.70) having the least closure. Nocturnal surface energy closure was very low at 0.11, and this was linked to low friction velocity during night-time, with results showing an increase in closure with increase in friction velocity.

The surface energy partitioning of this savanna ecosystem showed that sensible heat flux dominated the energy partitioning between March and October, followed by latent heat flux, and lastly the soil heat flux, and during the wet season where latent heat flux dominated the sensible heat flux. An increase in net radiation was characterized by an increase in both LE and H, with LE showing a higher rate of increase than H in the wet season, and the reverse happening during the dry season. An increase in VPD is characterized by a decrease in LE and increase in H during the wet season, and an increase of both fluxes during the dry season.

## 1. Introduction

Net solar radiation (Rn) reaching the earth's surface determines the amount of energy available for latent (LE), sensible (H) and soil (G) heat fluxes, and other minor fluxes such as heat stored by the canopy and the ground. Energy partitioning on the earth's surface is a function of interactions between biogeochemical cycling, plant physiology, the state of the atmospheric boundary layer and climate (Wilson et al., 2002). How the turbulent fluxes (H and LE) are partitioned in an ecosystem plays a critical role in determining the hydrological cycle, boundary layer development, weather and climate (Falge et al., 2005). Understanding the partitioning of energy, particularly the turbulent fluxes, is important for water resource management in (semi) arid regions, where potential evapotranspiration far exceeds precipitation.

Eddy covariance (EC) systems are currently the most reliable method for measuring carbon, energy and water fluxes, and they have become a standard technique in the study of surface-atmosphere boundary layer




interactions. They provide a distinct contribution to the study of environmental, biological and climatological
controls of the net surface exchanges between the land surface (including vegetation) and the atmosphere
(Aubinet, et al., 1999; Baldocchi et al., 2001). The accuracy of these data is very important because they are used
to validate and assess performance of land surface and climate models. However, the EC techniques have
limitations in terms of data processing and quality control methods, especially under complex conditions (e.g.,
unfavorable weather, such as high turbulence and low wind speed, and heterogeneous topography). In EC
measurements, the ideal situation is that available energy, i.e. net radiation minus soil heat flux is equal to the sum
of the turbulent fluxes (Rn-G = LE+H); however, in most instances, the measured available energy is larger than
the sum of the measurable turbulent fluxes of sensible heat and latent heat. Extensive research on the issue of
surface energy imbalance in EC observations has been done (Barr et al., 2012; Chen et al., 2009; Foken et al.,
2010; Franssen et al., 2010; Mauder et al., 2007), and closure error (or imbalance) has been documented to be
around 10-30 %. Causes for non-closure include unaccounted soil and canopy heat storage, non-inclusion of the
low and high frequency turbulence in the computation of the turbulent fluxes, land surface heterogeneities,
systematic measurement and sampling errors. This imbalance has implications on how energy flux measurements
should be interpreted and how these estimates should be compared with model simulations. The surface energy
balance closure is an accepted performance criterion of EC flux data (Twine et al., 2000; Wilson et al., 2002), and
different methods have been used to assess the energy closure and partitioning, including ordinary least squares
regression (OLS) method, i.e. a plot of turbulence fluxes (H+LE) against available energy (Rn-G), the residual
method, i.e. Rn-G-H-LE, and the energy balance ratio, i.e. LE+H/Rn-G.

Several researchers have investigated surface energy partitioning and energy balance closure for different

ecosystems, including savannas. Bagayoko et al. (2007) examined the seasonal variation of the energy balance in
West African savannas, and noted that latent heat flux played a major role in the wet season, whereas sensible
heat flux was significant in the dry season. In the grassland Mongolian Plateau, Li et al. (2006) concluded that
sensible heat flux dominated the energy partitioning, followed by ground heat flux, with the rainy season showing
slight increase in latent heat flux. Gu et al. (2006) used different ratios (Bowen ratio, G/Rn, H/Rn and LE/Rn) to
investigate surface energy exchange in the Tibetan Plateau, and showed that during the vegetation growth period,
LE was higher than H, and this was reversed during the post-growth period.

Research using the Skukuza EC system data has focused mainly on the carbon exchange, fire regimes, and

in global analysis of the energy balance (Archibald et al., 2009; Kutsch et al., 2008; Williams et al., 2009).
However, there has been no investigation of surface energy partitioning and energy balance closure in this
ecosystem. In this study, we examined the surface energy balance partitioning into soil heat conduction,
convection (sensible) and latent heat components and its energy balance closure using 15 years (2000-2014) of
eddy covariance data from the Skukuza flux tower.

First, a multi-year surface energy balance closure (EBC) analysis was done, including the seasonal and day-

night EBC evaluations, and an assessment of its error sources. This included investigating how friction velocity
affects the closure, and its link to low nighttime EBC. Surface energy partitioning during the wet and dry seasons
was examined, including how meteorological conditions such as vapour pressure deficit (VPD) and Rn affect the
partitioning.





## 2. Materials and methods

### 2.1. Site description

The Skukuza flux tower (25.02ºS, 31.50ºE) was established early 2000 as part of the SAFARI 2000 campaign and experiment, set up to understand the interactions between the atmosphere and the land surface in southern Africa by connecting ground data of carbon, water, and energy fluxes with remote sensing data generated by Earth observing satellites (Scholes et al., 2001; Shugart et al., 2004).

The site is located in the Kruger National Park (South Africa) at 365 m above sea level, and receives 550 ± 160 mm precipitation per annum between November and April, with significant inter-annual variability. The year is divided into a hot, wet growing season and a warm, dry non-growing season. The soils are generally shallow, with coarse sandy to sandy loam textures (about 65 % sand, 30 % clay and 5% silt). The area is characterised by a catenal pattern of soils and vegetation, with broad-leaved *Combretum* savanna on the crests dominated by the small trees (*Combretum apiculatum*), and fine-leaved *Acacia* savanna in the valleys dominated by *Acacia nigrescens* (Scholes et al., 1999). The vegetation is mainly open woodland, with approximately 30 % tree canopy cover of mixed *Acacia* and *Combretum* savanna types. Tree canopy height is 5–8 m with occasional trees (mostly *Sclerocarya birrea*) reaching 10 m. The grassy and herbaceous understory comprises grasses such as *Panicum maximum*, *Digitaria eriantha*, *Eragrostis rigidor*, and *Pogonarthria squarrosa*.

#### 2.1.1. Eddy covariance system

Since 2000, ecosystem-level fluxes of water, heat and carbon dioxide are measured using an eddy covariance system mounted at 16 m height of the 22 m high flux tower. The measurements taken and the instruments used are summarised in Table 1.

(Table 1)

From 2000 to 2005, H and LE were derived from a closed-path $CO_2$/$H_2O$ monitoring system, which was replaced by the open-path gas analyser in 2006. Also, from 2000 to 2008, incident and reflected shortwave radiation (i.e. 300–1100 nm, $Wm^{-2}$), incident and reflected near-infrared (600–1100 nm, $Wm^{-2}$) and incoming and emitted longwave radiation (>3.0 µm, $Wm^{-2}$) measurements were made using a two-component net radiometer (Model CNR 2: Kipp & Zonen, Delft, The Netherlands) at 20 s intervals and then recorded in the data-logger as 30 min averages; this was replaced with the Kipp & Zonen NRlite net radiometer in 2009. Soil heat flux is measured using the HFT3 plates (Campbell Scientific) installed at 5 cm below the surface at three locations, two under tree canopiesa and one between canopies.

Ancillary meteorological measurements include air temperature and relative humidity, also measured at 16 m height, using a Campbell Scientific HMP50 probe; precipitation at the top of the tower using a Texas TR525M tipping bucket rain gauge; wind speed and direction using a Climatronics Wind Sensor; and soil temperature using Campbell Scientific 107 soil temperature probe.

#### 2.1.2. Data pre-processing

Post-processing of the raw high frequency (10 Hz) data for calculation of half-hour periods of the turbulent fluxes and $CO_2$ ($F_c$; g $CO_2$ $m^{-2}$ $time^{-1}$) involved standard spike filtering, planar rotation of velocities and lag correction to $CO_2$ and q (Aubinet et al., 1999; Wilczak et al., 2001). All fluxes are reported as positive upward from the land to the atmosphere. Frequency response correction of some of the energy lost due to instrument separation, tube





attenuation, and gas analyzer response for LE and $F_c$ was performed with empirical co-spectral adjustment to
match the H co-spectrum (Eugster and Senn, 1995; Su et al., 2004).
**2.2.   Data analysis**
Half-hourly measurements of eddy covariance and climatological data from 2000 to 2014 were used to assess
surface energy partitioning and closure. Screening of the half-hourly data rejected i) data from periods of sensor
malfunction (i.e. when there was a faulty diagnostic signal), (ii) incomplete 30-minute datasets of Rn, G, LE and
H, and iii) outliers. After data screening, flux data with non-missing values of Rn, G, LE and H data were arranged
according to monthly and seasonal periods (summer (December – February), autumn (March – May), winter (June
– August), and spring (September – November)), as well as into daytime and nighttime.
**2.2.1.   Surface energy balance assessment**
The law of conservation of energy states that energy can neither be created nor destroyed, but is transformed from
one form to another, hence the ideal surface energy balance equation is written as:
$$Rn - G = H + LE \qquad\qquad\qquad (1)$$
Energy imbalance occurs when both sides of the equation do not balance. The energy balance closure was
evaluated at different levels, i.e. multi-year, seasonal, and day/ night periods (the assumption being that daytime
has positive Rn and nighttime has negative Rn),  using two methods, i.e.
i)       The ordinary least squares method (OLS), which is the regression between turbulent fluxes  and available
energy.
Ideal closure is when the intercept is zero and slope and the coefficient of determination are one. An assumption
is made using this method, that there are no random errors in the independent variables, i.e. Rn and G, which of
course is an incorrect assumption.
ii)      The energy balance ratio (EBR), which is ratio of the sum of turbulent fluxes to the available energy,
$\sum(LE + H)/\sum(Rn - G)$.
The EBR gives an overall evaluation of energy balance closure at longer time scales by averaging over errors in
the half-hour measurements; and the ideal closure is 1. EBR has the potential to remove biases in the half-hourly
data, such as the tendency to overestimate positive fluxes during the day and underestimate negative fluxes at
night.
To investigate the effect of friction velocity on EBR and how it is related to time of day, using friction
velocity, the data were separated into 4 25-percentiles, and the EBR and OLS evaluated.
**2.2.2.   Analyzing surface energy partitioning**
To evaluate solar radiation variation and partitioning into latent and sensible heat fluxes in this biome, EC surface
energy data from 2000 to 2014 were used. The data gaps in these data were first filled using the Amelia II software
(Honaker et al., 2011). This R-program was designed to impute missing data using a bootstrapping-based multiple
imputation algorithm. The minimum, maximum and mean statistics of Rn, H, LE and G were then estimated.
The monthly and seasonal variations of energy partitioning were assessed. Surface energy partitioning
was also characterized as a direct function of vapor pressure deficit (VPD) and Rn during the wet and dry seasons.





## 3. Results and discussion

### 3.1. Meteorological conditions

Fig 1 shows the 15-year average daily temperature, VPD and rainfall totals at the Skukuza flux tower. The annual average temperatures over the 15-year period ranged between 21.13°C in 2012 and 23.23 °C in 2003, with a 15-year average temperature of 22.9 °C. While 2003 was the hottest year, it was also the driest year, with annual rainfall of 273.6 mm, with 2002 also recording very low rainfall of 325.4 mm, both receiving rainfall amounts below the recorded mean annual rainfall of 550±160 mm. The wettest years were 2013, 2000, 2014 and 2004 which received 1414, 1115.6, 1010.2 and 1005.7 mm, respectively. 2007 and 2008 had incomplete rainfall data records to assess their annuals. The annual daily average VPD was between 0.024 and 4.03 kPa, with an overall average of 1.28 ± 0.62 kPa. The daily average VPD decreased with rainy days, and showed an increase during rain-free days. The wet years, i.e. 2000, 2013 and 2014 had low annual average VPD of 1.98, 1.34 and 1.83 kPa, respectively, whereas the drought years exhibited high VPDs with 2002 and 2003 with 2.77 and 2.97 kPa, respectively. The long-term weather records are comparable with the 1912 – 2001 and 1960 – 1999 climate analysis for the same area as reported by Kruger et al. (2002) and Scholes et al. (2001), showing a mean annual total precipitation of 547.1 mm and air temperature of 21.9 ºC. The low rainfall during 2000-2003 seasons was also reported by Kutch et al. (2008), who were investigating the connection between water relations and carbon fluxes during the mentioned period.

**(Figure 1)**

### 3.2. Surface energy balance assessment

Data completeness varied largely 3.24 % (2013) and 57.65 % (2010), with a mean of 30.77 %. The variation in data completeness is due to a number of factors including instrument failures, changes and (re)calibration, and poor weather conditions.

#### 3.2.1. Multi-year analysis of surface energy balance closure

Fig 2 summarizes results of the multi-year energy balance closure analysis for the Skukuza eddy covariance system from 2000 to 2014. The slopes ranged between 0.93 and 1.47, with a mean 1.19 with standard deviation of 0.21, and the intercepts were a mean of 17.79 with standard deviation of 32.96 Wm$^{-2}$. $R^2$ ranged between 0.73 in 2005 and 0.92 in 2003, with a mean of 0.86 with standard deviation of 0.05.

The annual energy balance ratio (EBR) for the 15 years ranged between 0.44 in 2007 and an extreme 3.76 in 2013, with a mean of 0.97±0.81. Between 2004 and 2008, EBR ranges between 0.44 and 0.53, whereas from 2000 to 2003 and 2009 to 2014, the EBR ranged 0.76 and 1.09. The EBR for 2010 to 2012 were slightly greater than 1, indicating an overestimation of the turbulent fluxes (H+LE) compared to the available energy. The remaining years were less than 1, indicating that the turbulent fluxes were lower than the available energy. The period of low EBR between 2004 and 2008 is characterized by the absence of negative values of available energy (Rn-G) as illustrated in Fig 2. Between 2000 and 2004, the CNR2 net radiometer was used to measure long and shortwave radiation, and these were combined to derive Rn. However, when the pyrgeometer broke down in 2004, Rn was derived from measured shortwave radiation and modelled longwave radiation until the CNR2 was replaced by the NRLite net radiometer in 2009. This was a significant source of error, as shown by the low EBR between 2004 and 2008. The closed-path gas analyzer was also changed to open-path gas analyzer in 2006. An





analysis of the 2006 data (which had very low data completeness of 7.59 %) showed that there were no
measurements recorded until September, possibly due to instrument failure. Our final proposed mean multiyear
EBR estimate for the 15-year period (2000-2014), excluding the years with data issues (2004 to 2008, and 2013),
was therefore 0.93 ± 0.11. For further analysis of the EBR, we excluded the years with bad data.
**(Figure 2)**
The EBR results for the Skukuza eddy covariance system, with a mean of 0.93 (only the years with good data),
are generally within the reported accuracies by most studies that report the energy balance closure error at 10 –
30%. Chen et al. (2009) report a mean of 0.98 EBR, average slope of 0.83, and $R^2$ ranges between 0.87 and 0.94
for their study in the semi-arid region of Mongolia. Wilson et al., (2002) also reported that the mean annual EBR
for 22 FLUXNET sites was 0.84, ranging from 0.34 to 1.69, and slopes and intercepts ranging from 0.53 to 0.99,
and from −33 to 37 W m$^{-2}$, respectively. Yuling et al. (2005) also report that in the ChinaFLUX, EBR ranged
between 0.58 and 1.00, with a mean of 0.83. von Randow et al. (2004) showed an energy imbalance of 26 % even
after correcting for the angle of attack on the sonic anemometer in the forested Jeru study area in the Amazon,
and explained this as due to either slow wind direction changes which result in low frequency components that
cannot be captured using short time rotation scales, and the difficulty in estimating horizontal flux divergences
caused by energy that is transported horizontally by circulations. Sanchez et al., (2010) showed that the inclusion
of the storage term in the EBR improved the closure by almost 6 % from 0.72, in their study in a FLUXNET
boreal site in Finland.  Using data from the Tibetan Observation and Research Platform (TORP), Liu et al. (2011)
observed an EBR value of 0.85 in an alfalfa field in semi-arid China. Also under similar semi-arid conditions, in
China, an EBR value of 0.80 was found by Xin and Liu (2010) in a maize crop. Were et al. (2007) reported EBR
values of about 0.90 over shrub and herbaceous patches, in a dry valley in southeast Spain.

### 3.2.2.    Seasonal variation of EBR

Fig 3 shows the seasonal OLS results for the 15 year period, excluding years 2004 to 2008 and 2013. The slopes
ranged between 0.94 and 1.21, with a mean of 1.10 ± 0.11, and the intercepts were a mean of 11.97 Wm$^{-2}$ ± 3.87
Wm$^{-2}$. $R^2$ ranged between 0.74 and 0.88 with a mean of 0.83 ±0.06. The EBR for the different seasons ranged
between 0.70 and 1.02, with a mean of 0.88 ± 0.14. The winter season had the lowest EBR of 0.70, while summer
and spring were closest to unity with EBR of 0.98 and 1.02, respectively, and autumn had EBR of 0.84. A large
number of outliers is observed in summer due to cloudy weather conditions and rainfall events that make the
thermopile surface wet, thus reducing the accuracy of the net radiometer. A study comparing different the
performance of different net radiometers by Blonquist et al. (2009) shows that the NR-Lite is highly sensitive to
precipitation and dew/ frost since it the sensor is not protected.
**(Figure 3)**
Wilson et al. (2002) comprehensively investigated the energy closure of the summer and winter seasons for 22
FLUXNET sites for 50 site-years. They also reported higher energy balance correlation during the wet compared
to the dry season, with the mean $R^2$ of 0.89 and 0.68, respectively. However, their EBR showed smaller differences
between the two seasons, being 0.81 and 0.72, for summer and winter, respectively, whereas for Skukuza, the
differences were much significant. Ma et al. (2009) reported an opposite result from the Skukuza results, showing
energy closures of 0.70 in summer and 0.92 in winter over the flat prairie on the northern Tibetan Plateau.





### 3.2.3. Day – night-time effects

Fig 4 shows the daytime and nocturnal OLS regression results for the 15 year period. The daytime and nocturnal slopes were 0.99 and 0.11, with the intercepts being 76.76 and 1.74 $Wm^{-2}$, respectively. Daytime and nocturnal $R^2$ were 0.64 and 0.01, respectively. The EBR for the different times of day were 0.96 and 0.27, daytime and nocturnal, respectively.

**(Figure 4)**

Other studies also reported a higher daytime surface energy balance closure. For instance, Wilson et al., (2002) showed that the mean annual daytime EBR was 0.8, whereas the nocturnal EBR was reported to be was negative or was much less or much greater than 1.

To understand the effect of friction velocity on the energy balance closure, which had friction velocity ($u_*$) data, were used. Using friction velocity, the data were separated into 4 25-percentiles, and the EBR and OLS evaluated. Results show that the first quartile, the EBR was 3.94, with the 50-percentile at 0.99, the third quartile at unity, and the fourth quartile at 1.03 (Fig 5). The slopes were between 1.01 and 1.12, with the intercepts ranging between -9.26 and -0.17 $Wm^{-2}$, whereas $R^2$ were 0.82, 0.86, 0.85 and 0.81 for the first to the fourth quartiles, respectively.

**(Figure 5)**

A quick assessment shows that the time associated with the low friction velocities, i.e. the first quartile are night-time data constituting 81 % of the whole first quartile dataset, and the last quartile had the highest number of daytime values at 79.29 % of the fourth quartile dataset. Lee and Hu (2002) hypothesized that the lack of energy balance closure during nocturnal periods was often the result of mean vertical advection, whereas Aubinet et al., (1999) and Blanken et al., (1997) showed that energy imbalance during nocturnal periods is usually greatest when friction velocity is small. Another source of error in the nocturnal EBR is the high uncertainty in night-time measurements of Rn. At night, the assumption is that there is no shortwave radiation, and Rn is a product of longwave radiation. Studies show that night-time measurements of longwave radiation were less accurate than daytime measurements (Blonquist et al., 2009). The RN-Lite, for instance has low sensitivity to longwave radiation, resulting in low accuracy in low measurements.

### 3.3. Surface energy partitioning

#### 3.3.1. Surface energy measurements

The mean daily and annual measurements of the energy budget components from 2000 to 2014 are highlighted in Fig 6 and Table 2. The seasonal cycle of each component can be seen throughout the years, where at the beginning of each year the energy budget components are high, and as each year progresses they all decrease to reach a low during the middle of the year, which is the winter/ dry season, and a gradual increase being experienced during spring right to the summer at the end of each year. The multi-year daily means of Rn, H, LE and G were 139.1 $Wm^{-2}$, 57.70 $Wm^{-2}$, 42.81 $Wm^{-2}$ and 2.94 $Wm^{-2}$, with standard deviations of 239.75 $Wm^{-2}$, 104.15 $Wm^{-2}$, 70.58 $Wm^{-2}$ and 53.67 $Wm^{-2}$, respectively.

**(Figure 6)**

The gaps in 2006 and 2013 indicates the absence of the surface energy flux measurements in those years, which was a result of instrument failure. Between 2004 and 2008, the Rn was calculated as a product of measured





shortwave radiation and modelled longwave radiation, which was a high source of error in the estimation of Rn.
These years are also characterised by poor energy balance closure, as shown in Section 3.2.1 above.
(**Table 2**)

### 3.3.2.    Influence of weather conditions and seasonality
In arid/semi-arid ecosystems, solar radiation is not a limiting factor for evapotranspiration, instead it is mainly
limited by water availability. The seasonal fluctuations of energy fluxes are affected by the seasonal changes in
the solar radiation, air temperature, precipitation and soil moisture (Baldocchi et al., 2000; Arain et al., 2003).
These climatic variables influence vegetation dynamics in an ecosystem, as well as how solar radiation is
partitioned. Hence, daily measurements of precipitation, air temperature and VPD were evaluated to investigate
the partitioning of the surface energy in the semi-arid savanna landscape of Skukuza.)
(**Figure 7**)

To illustrate the partitioning of solar radiation into the different fluxes throughout the year, Fig 7 presents

the multi-year mean monthly variations of the surface energy components showing a general decrease of the
components between February and June, which then gradually increases again until November. The multi-year
monthly means of Rn, H, LE and G were 71.27 $Wm^{-2}$ (June) and 197.33 $Wm^{-2}$ (November), 37.11 $Wm^{-2}$ (June)
and 80.37 $Wm^{-2}$ (November), 8.52 $Wm^{-2}$ (August) and 127.17 $Wm^{-2}$ (December), -2.28 $Wm^{-2}$ (June) and 20.78
$Wm^{-2}$ (November), respectively. The month of August had the highest BR of 6.42, whereas December had the
least at 0.42. The residual accounted for between -19.69 and 34.74 % of Rn, and an average of 4.70 %.

The general trend shows that sensible heat flux dominated the energy partitioning between May and

October, followed by latent heat flux, and lastly the soil heat flux, except during the wet season where latent heat
flux was larger than sensible heat flux. This is illustrated by the trend of BR, showing  an increase of BR from
April, with the peak in August, then a steady decrease until it hits lowest in December. The period of low BR is
characterised by high incoming solar radiation, with high Rn and high precipitation (Fig 1). As the season
transitions into winter, it is characterised by reduced net radiation and low measurements H and LE.

Just before the first rains, i.e. between September and November, tree flowering and leaf emergence

occurs in the semi-arid savanna in the Skukuza area (Archibald and Scholes, 2007), and grasses shoot as soil
moisture availability improves with the rains (Scholes et al., 2003). This is characterised by a gradual increase in
latent heat flux (evapotranspiration) and decrease in BR, which, when compared to the winter season, is
significantly lower than the sensible heat flux, as illustrated in Fig 7. As the rainy season progresses, and
vegetation development peaks, latent heat flux also reaches its maximum, becoming significantly higher than
sensible heat flux, and hence, high BR. Between March and September, when leaf senescence occurs, the leaves
gradually change colour to brown and grass to straw, and trees defoliate, sensible heat flux again gradually
becomes significantly higher than LE.

The influence of VPD and Rn on surface energy partitioning was investigated during the wet and dry

seasons. Results show that there is an increase in H and decrease in LE with an increase in VPD in the wet season
(Fig 9). As illustrated earlier (Fig 1), VPD is higher when there is little or no rain (low soil water availability),
which explains the increase in H with a rise VPD (Fig 9d). In this instance, although the evaporative demand is
high, the stomatal conductance is reduced due to absence of water in the soil, resulting in smaller LE and higher
H. Rn, on the other hand, is partitioned into different fluxes, based on other climatic and vegetation physiological





characteristics. Figure 10 illustrates that both latent and sensible heat flux increase with increase in net radiation,
although their increases are not in proportion, based on season. During the wet season, the rate of increase of LE
is higher than that of H, whereas in the dry season the reverse is true. The rate of increase of LE is controlled by
the availability of soil water (precipitation), (also illustrated in Figures 1c and 6 (LE)), and during the wet season
it increases steadily with increasing Rn, whereas the rate of increase of H is concave, showing saturation with an
increase in Rn. The opposite is true during the dry season, with limited water availability, the rate of increase of
LE slows down with increase in Rn, and a steady increase of H with Rn increase.
Gu et al. (2006) examined how soil moisture, vapor pressure deficit (VPD) and net radiation control
surface energy partitioning at a temperate deciduous forest site in central Missouri, USA. They ascertained that
with ample soil moisture, latent heat flux dominates over sensible heat flux, and reduced soil moisture availability
reversed the dominance of latent heat over sensible heat, because of its direct effect on stomatal conductance. An
increase in net radiation, on the other hand, also increases both sensible and latent heat fluxes. The increase of
either then becomes a function of soil moisture availability, since they cannot increase in the same proportion.
Their findings are generally consistent with our results, which show that during the rainy season, latent heat flux
was significantly higher than sensible heat flux, whereas, during the other seasons, sensible heat flux remained
higher than latent heat flux. However, their findings on the effect of VPD on energy partitioning are opposite of
our study, where they record significant increase in LE and decrease in H with a rise in VPD during the non-
drought period; in dry conditions, both components show slight increases with increase in VPD. Li et al. (2012)
also investigated the partitioning of surface energy in the grazing lands of Mongolia, and concluded that the energy
partitioning was also controlled by vegetation dynamics and soil moisture availability, although soil heat flux is
reportedly higher than latent heat flux in most instances. In a temperate mountain grassland in Austria, Harmmerle
et al., (2008) found that the energy partitioning in this climatic region was dominated by latent heat flux, followed
by sensible heat flux and lastly soil heat flux.
The consensus in all above studies is that vegetation and climate dynamics play a critical role in energy
partitioning. They note that during full vegetation cover, latent heat flux is the dominant portion of net radiation.
However, depending on the climatic region, the limiting factors of energy partitioning vary between water
availability and radiation. Our study confirms that in semi-arid regions, sensible heat flux is the highest fraction
of net radiation throughout the year, except during the wet period, when latent heat flux surpasses sensible heat
flux. However, in regions and locations where water availability is not a limiting factor, latent heat flux may take
the highest portion of net radiation.

## 4. Conclusion

This study investigated both surface energy balance and its partitioning into turbulent fluxes during the wet and
dry seasons in a semi-arid savanna ecosystem in Skukuza using eddy covariance data from 2000 to 2014. The
analysis revealed a mean multi-year energy balance ratio of 0.93, The variation of RBR based on season, time of
day and as a function of friction velocity was explored. The seasonal EBR varied between 0.70 and 0.88, with
winter recording the highest energy imbalance. Daytime EBR was as high as 0.96, with 0.27 EBR for the
nighttime. The high energy imbalance at night was explained as a result of stable conditions, which limit
turbulence that is essential for the creation of eddies. The assessment of the effect of friction velocity on EBR



showed that EBR increased with an increase in friction velocity, with low friction velocity experienced mainly
during night-time.
The energy partition analysis revealed that sensible heat flux is the dominant portion of net radiation in
this semi-arid region, except in summer, when there is rainfall. The results also show that water availability and
vegetation dynamics play a critical role in energy partitioning, whereby when it rains, vegetation growth occurs,
leading to an increase in latent heat flux / evapotranspiration. Clearly an increase in Rn results in a rise in H and
LE, however their increases are controlled by water availability. During the wet season, the rate of increase of LE
is higher than that of H, whereas in the dry season the reverse is true. The rate of increase of LE is controlled by
the availability of soil water (precipitation), and during the wet season it increases steadily with increasing Rn,
whereas the rate of increase of H shows saturation with an increase in Rn. The opposite is true during the dry
season, with limited water availability, the rate of increase of LE reaches saturation with increase in Rn and a
steady increase of H with Rn increase. An increase in VPD, on the other hand, results in an increase in H and
decrease in LE, with higher VPD experienced during the dry season, which explains the high H, although the
evaporative demand is high.

**Acknowledgements**
This study was supported by the Council for Scientific and Industrial Research under the project entitled
"Monitoring of water availability using geo-spatial data and earth observations", and the National Research
Foundation under the Thuthuka PhD cycle grant.

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





**Table 1: Measurements taken and instruments used at Skukuza flux tower**

| Instrument | Model/ brand | Measurement |
|---|---|---|
| Sonic anemometer | Gill Instruments Solent R3, Hampshire, England | 3-dimensional, orthogonal components of velocity (u, v, w (ms$^{-1}$)) |
| Closed path gas analyser | IRGA, LiCOR 6262, LiCOR, Lincoln | Water vapor, carbon dioxide concentrations |
| Radiometer | Kipp and Zonen CNR1, Delft, The Netherlands | Incoming and outgoing longwave and shortwave radiation |
| HFT3 plates | Campbell Scientific | Soil heat flux at 5 cm depth with 3 replicates, i.e. two under tree canopies and one on open space |
| Frequency domain reflectometry probes | Campbell Scientific CS615, Logan, Utah | Volumetric soil moisture content with two in the clayey Acacia – dominated soils downhill of the tower at 3, 7, 16, 30, and 50 cm, and another two at 5, 13, 29, and 61 cm in the sandier Combretum – dominated soils uphill |







**Table 2: Statistical summary of annual values of the energy balance components**

| Year | % data completion | | H | LE | G | Rn |
|---|---|---|---|---|---|---|
| 2000 | 14.16 | Max | 470.31 | 422.89 | 191.53 | 817.60 |
| | | Min | -139.77 | -72.43 | -61.60 | -95.93 |
| | | Mean | 45.82 | 36.11 | 5.32 | 91.46 |
| 2001 | 12.78 | Max | 790.82 | 513.09 | 292.87 | 899.90 |
| | | Min | -159.87 | -85.95 | -90.27 | -116.58 |
| | | Mean | 58.56 | 43.68 | 9.27 | 128.27 |
| 2002 | 17.77 | Max | 415.93 | 174.07 | 171.93 | 583.30 |
| | | Min | -117.66 | -89.16 | -86.00 | -122.21 |
| | | Mean | 61.35 | 10.29 | 4.10 | 90.72 |
| 2003 | 41.50 | Max | 556.21 | 308.71 | 217.60 | 879.30 |
| | | Min | -92.99 | -97.81 | -106.23 | -116.04 |
| | | Mean | 58.15 | 21.68 | 6.17 | 94.53 |
| 2004 | 28.21 | Max | 505.36 | 498.10 | 129.96 | 925.30 |
| | | Min | -150.08 | -89.07 | -69.76 | -5.88 |
| | | Mean | 56.46 | 17.99 | 7.97 | 156.10 |
| 2005 | 35.37 | Max | 606.28 | 737.43 | 288.20 | 933.20 |
| | | Min | -130.40 | -97.00 | -107.37 | -4.92 |
| | | Mean | 51.43 | 17.82 | 0.99 | 159.09 |
| 2006 | 7.59 | Max | 583.66 | 331.25 | 335.30 | 1003.30 |
| | | Min | -72.45 | -119.09 | -72.80 | -6.56 |
| | | Mean | 84.67 | 35.94 | 19.69 | 247.70 |
| 2007 | 48.77 | Max | 552.93 | 426.34 | 340.67 | 1011.30 |
| | | Min | -131.40 | -130.79 | -129.70 | -6.71 |
| | | Mean | 59.04 | 14.32 | 4.14 | 169.84 |
| 2008 | 54.30 | Max | 616.43 | 439.76 | 238.57 | 1038.50 |
| | | Min | -140.13 | -144.97 | -104.60 | -5.91 |
| | | Mean | 63.06 | 26.30 | 6.22 | 191.26 |
| 2009 | 42.69 | Max | 551.34 | 776.62 | 328.93 | 1060.50 |
| | | Min | -96.68 | -135.43 | -94.20 | -155.90 |
| | | Mean | 55.42 | 96.54 | 6.87 | 207.77 |
| 2010 | 57.65 | Max | 626.68 | 624.38 | 199.33 | 888.00 |
| | | Min | -173.11 | -135.62 | -66.35 | -180.70 |
| | | Mean | 57.23 | 52.54 | 3.74 | 105.10 |
| 2011 | 41.34 | Max | 591.16 | 688.46 | 171.27 | 832.00 |
| | | Min | -135.77 | -127.02 | -58.59 | -96.50 |
| | | Mean | 63.88 | 73.11 | 1.75 | 127.94 |
| 2012 | 27.62 | Max | 572.11 | 566.88 | 185.80 | 899.00 |
| | | Min | -171.83 | -148.49 | -50.92 | -99.69 |
| | | Mean | 59.25 | 52.49 | 2.16 | 111.31 |
| 2013 | 3.25 | Max | 317.98 | 661.09 | 79.67 | 742.05 |
| | | Min | -62.96 | -27.19 | -30.49 | -90.30 |
| | | Mean | 1.79 | 34.08 | -15.64 | -6.09 |
| 2014 | 28.66 | Max | 533.46 | 726.31 | 89.50 | 893.00 |
| | | Min | -238.65 | -134.39 | -33.36 | -89.70 |
| | | Mean | 59.37 | 69.55 | 1.18 | 147.30 |






**Figures**

**Figures**

Figure 1: summaries of daily (a) average air temperature, (b) average VPD, and (c) total rainfall from 2000 to 2014







**Figure 2: 15-year series of annual regression analysis of turbulent (sensible and latent) heat fluxes against available**
**energy (net radiation minus ground conduction heat) from 2000 to 2014 at Skukuza, (SA). The colour bars represent**
**the count of EBR values.**





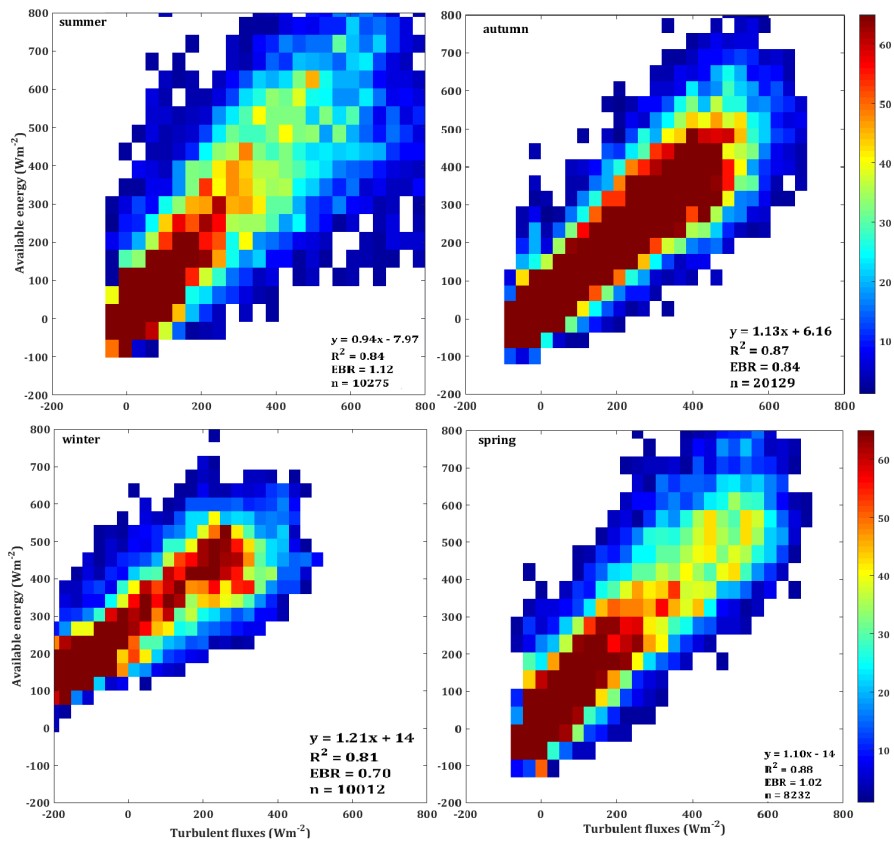

**Figure 3: Seasonal turbulent fluxes (H+LE) correlation to available energy (Rn-G) for Skukuza flux tower from summer(Dec-Feb), autumn (March-May), winter (June-Aug), spring (Sept-Nov). The colour bars represent the count of EBR values**





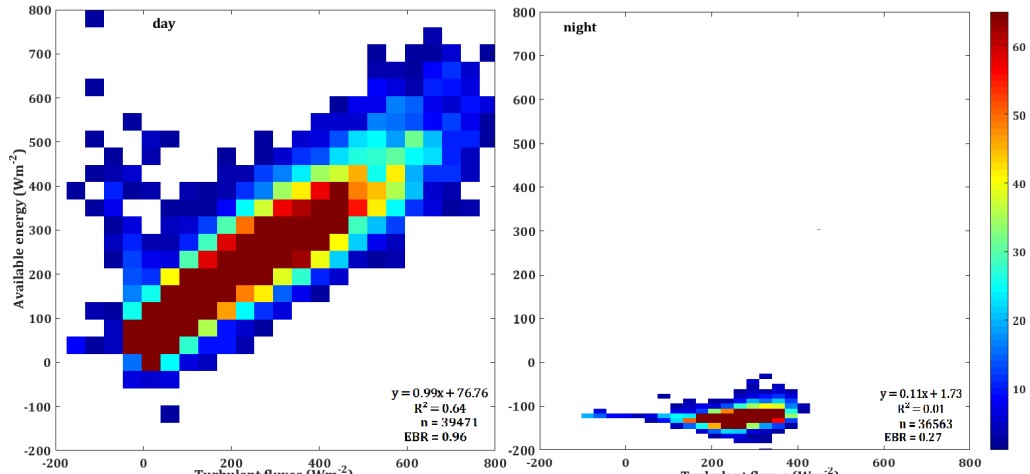

**Figure 4: Turbulent fluxes correlation to available energy for daytime (a) and night-time (b), using the full (2000-2014) 15-year available data series. The colour bars represent the count of EBR values**





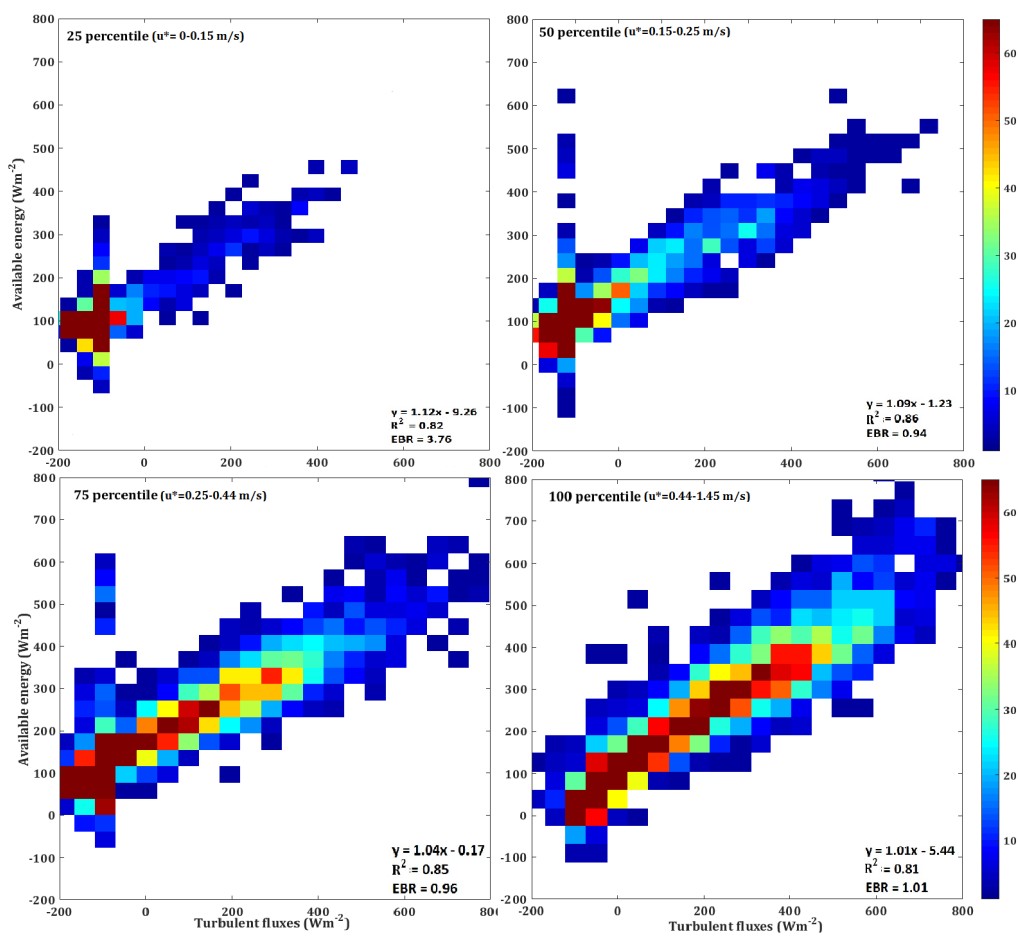

**Figure 5: OLS and EBR evaluations at different friction velocity sorted at 4 quartiles. The colour bar represents the**
**count of EBR values. The colour bars represent the count of EBR values.**

513  .






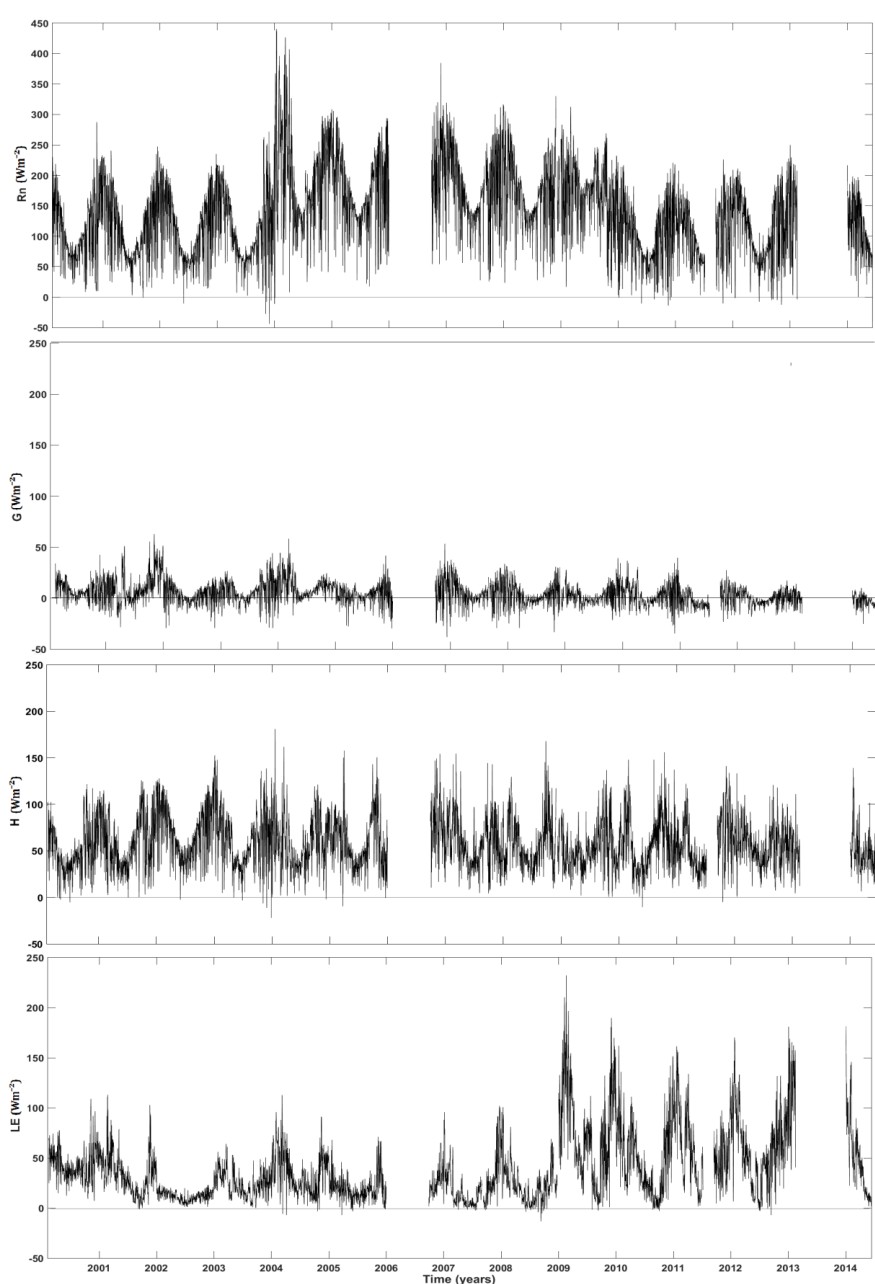

**Figure 6: Time series of daily mean surface energy balance component fluxes from 2000 to 2014 at Skukuza flux tower**
**site (SA)**








**Figure 7: 15-year (2000-2014) monthly means of surface energy balance fluxes of Skukuza flux tower site (SA),**
**highlighting the partitioning of Rn**





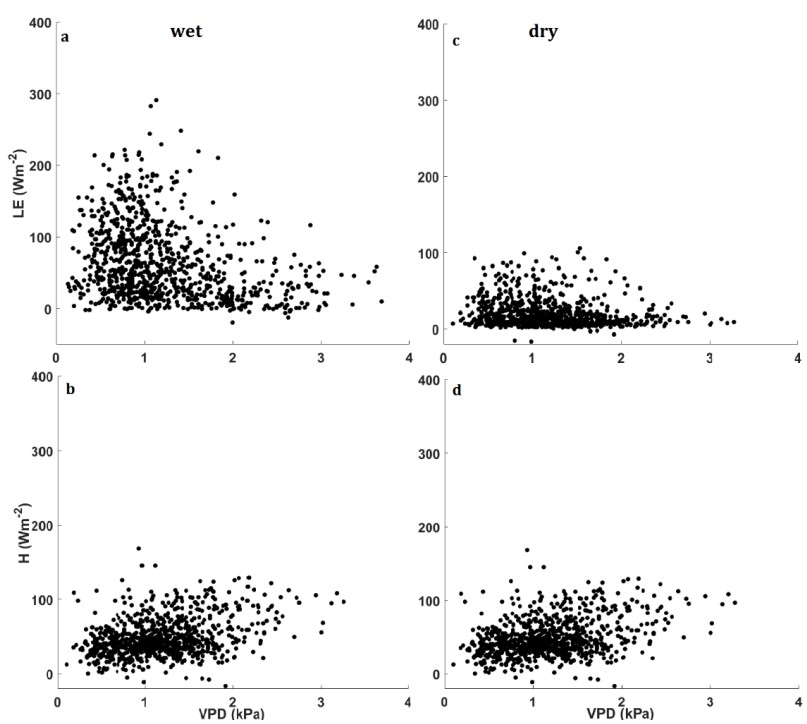


**Figure 8: Relationship between the fluxes and VPD under wet and dry conditions**




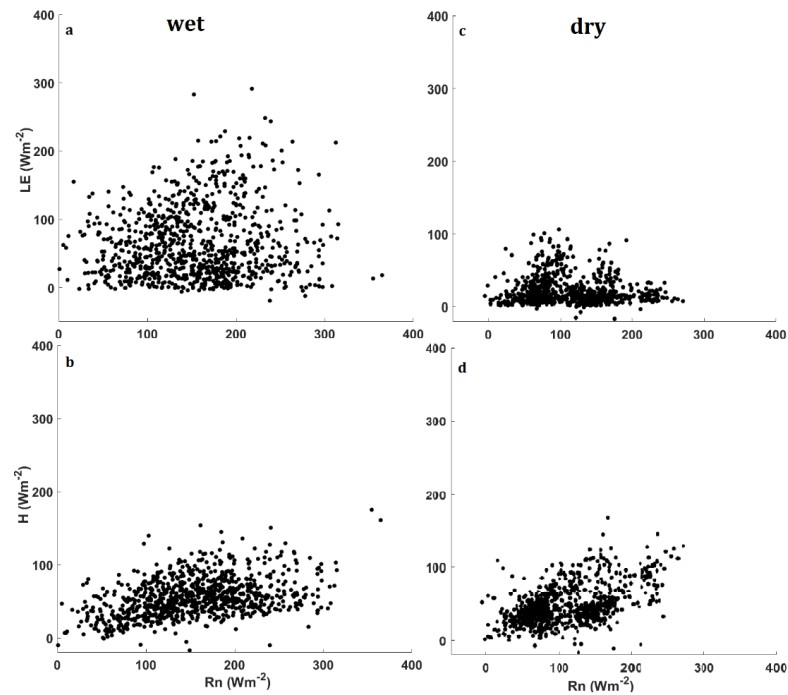

**Figure 9: Effects of net radiation on LE and H under wet and dry conditions**