# Peer review of "Analysing surface energy balance closure and partitioning 2 over a semi-arid savanna FLUXNET site in Skukuza, Kruger 3 National Park, South Africa"

_Hydrology and Earth System Sciences, 2016_

## Referee Comment (RC1) · N. van de Giesen (Referee) · 4 Feb 2017

General The article presents an impressive multi-year dataset of energy fluxes over an undersampled part of the world. The focus is on the energy balance closure.

Major remarks The article points out the difficulties of collecting valid data over long periods of time. My first question is if the cleaning procedures may have introduced any biases? For example, periods with rainfall often produce problems with sonic anemometers. Could you comment on this?

The second is a pet peeve of mine and concerns the ground heat flux. If I understand

correctly (the text is not so clear, see also below under minor remarks), you report EBR per half hour. Over the period of half an hour, ground heat flux can typically play an important role. Ground heat flux is also not very well captured with ground heatflux plates, which basically measure the temperature difference between the top and bottom of a piece of plastic in the ground. Even if the plates would work as intended, they are clearly biased as $\frac{2}{3}$ of the plates are under canopies while only 30% of the area has a canopy. I don't ask for you to go back and repeat the measurements with better measurements of G but a critical discussion is needed. A simple way to get some idea is to compare half hourly results with daily averaged EBRs. G will generally be negligible at daily scales while it can easily make up 50% of the energy balance at a half hourly basis.

Finally, the article would become ten times more valuable if you make the (cleaned?) dataset available online.

Minor remarks Line 29: Winter & summer are not so obvious terms for people not familiar with Kruger National Park. Either use months or, my preference, talk about dry and wet, as you do later under 2.1. Line 36: characterized by or rather correlated with? Line 41: Is the heat stored in the ground not the ground heat flux G? Line 47: Potential evapotranspiration is a problematic term. Better use "reference evaporation". Line 58: "measured" instead of "measurable" Line 92: Here you use Earth, elsewhere earth. I have no preference but best stick to one. Line 117: canopies instead of canopiesa Line 125: Did you use any software? Is code available? Line 126: You state that all upward fluxes are positive but later you clearly change this in Equation 1 and also when you state that daytime Rn is positive. Line 157: I surmised that you evaluated the dataset by looking at half hourly EBRs. The text here is, however, not very clear on that. Please make explicit. Line 198: Is the 0.81 the standard deviation in the estimate of the mean? Or is it the standard deviation? Also, with EBR always being larger than zero, perfect at one, and not upwardly bounded, would a logarithmic averaging scheme not make more sense? Lines 213 and further and in general throughout this part: You mix

literature review with results. It is more common not to introduce too much additional information from outside the study past the introduction. Would probably be better to move this to intro (but don't make it too long!). Line 230 and further: The Results and discussion focus on EBR and other outcomes in a very descriptive way. Would be better to already include more physical insights here as to why you see what you see. Line 264: Why the hurry? Here also please expand on role of G as mentioned above. Lines 334 and further: In general, there is a bit of a mix between the focus on EBR and the more general and the probably more interesting general interpretation of results. The article is built up around EBR and only towards the end do general energy & water availability considerations come up. Perhaps point to these earlier in the text. In any case, please shift the perspective from starting with other studies, such as by Gu et al., and comparing those with your results to a perspective that starts with your results and then compares those, preferably a bit more systematically, with other studies.

---

## Referee Comment (RC2) · Anonymous Referee #2 · 7 Feb 2017

General comments The authors evaluated a 15-year EC data record of a savanna FLUXNET site in Kruger National Park. This is a great and unique dataset. The authors focus in their analysis on the surface energy balance closure and energy partitioning. The topic fits very well into the scope of HESS, and the dataset will be interesting for a broad readership of HESS. The dataset is carefully evaluated for several aspect. The authors give interesting insight to technical problems that showed up over the 15 years, and they analyzed, among others, the effect of the season as well as the friction velocity on the EBR.

[Figure]

My major concern is related to the measurement of the ground heat flux. Firstly, important information is missing. As far as I got it, the authors did not determine the heat storage change in the layer above the heat flux plate (HFP). Please state this clearly in the Material and Method (MM) part. Moreover, it remains unclear how the three HFP readings were averaged. Two were installed under tree canopies and one at open space. How did you compute the mean ground heat flux representative for the footprint? Did you compute a weighted mean or did you compute simply the mean over the three plates. In the latter case the ground heat flux would be systematically underestimated, because the areal fraction of tree canopy is only 30%. Secondly, in general neglecting the heat storage term must result to a certain extent in a systematic underestimation of the ground heat flux and hence to a systematic overestimation of the available energy and consequently to an underestimation of the EBR. From own measurements (HFP were installed in 8 cm depth) I know that this storage term can reach at unshaded surfaces 50 to 100 W/m2. Please discuss in detail the magnitude of error that might originate from your methodological approach.

Moreover, I wondered why the authors do not give any information on monthly or annual evapotranspiration rates (in mm). With that information one could get a guess of the climatic water balance of that ecosystem. I think this would be very interesting for the reader and would further strengthen the manuscript.

Specific comments Line 132: Please state which software tool (e.g. TK3 or EddyPro) was used to process the EC raw data. Line 135: How did you detect outliers? Please explain! Line 170: The intention of Figure 1 is to show temperature, VPD and rainfall anomalies between the years. I think this way of displaying the data is not really optimal for this purpose. The authors should think about a better way to present these anomalies. One way could be to compute for every month the difference from the 15-year mean and list these differences in a table (rows: month; column: years). Months, for example, that were warmer than the 15-year average get a red color, months that were colder get a blue color. The larger the difference the more intensive the red and

blue color. Line 198: This is a little bit data cosmetic. The very good EBR is achieved thanks to the really bad year 2013, which had an EBR of 3.76. If you remove this year as outlier the mean EBR reduces to 0.77. I suggest that the authors start this chapter with explaining the technical problems that showed up over the years with the very low EBR and the extremely high EBR in 2013. And after that the authors should refer only to the years with no data or technical issues. Figure 2: In the OLS approach, the dependent variable (turbulent fluxes) is plotted against the independently derived available energy. See for example Wilson et al. (2002). If you plot it the other way round, as you did, the slope of the regression does not fit to the EBR. If the EBR is below one than the slope must also be below one. In the year 2007, for example, the EBR is 0.44 but the slope is 1.46. That does not fit together. Moreover, if you use the turbulent flux in a regression as independent variable your statistical model assumes that this variable has no error. Please correct everywhere the figures and update the numbers for slopes and intercepts! Line 205: Here it would be important for the reader to know how you modeled the in- and outcoming longwave radiation, so that they can avoid this mistake in future. Please describe this model in more detail.

Technical comments Line 28: Avoid the wording bad and good data. Please use instead e.g. low- and high quality data Line 40: I would not count energy stored in ground as a minor flux term (see above). Please rephrase. Line 83: If you start the sentence with first I expect that there comes a second item. Line 150: Replace "incorrect assumption" with "simplification". Line 148: Introduce here the symbol "$R^2$". Line 158: Rewrite "4" in "four". Line 224: Here it is unclear which storage term was included by Sanchez et al. (2010). Please rewrite! Table 1: Why clayey? In the MM part you write that the texture ranged from sand to loamy sand. Please check! Table 1: Campbell Scientific is not the manufacturer of the HFP. The manufacturer is Huskeflux. Please correct that and mention whether you used self-calibrating plates or not. Table 1: Beside the wind speed the anemometer measures also the sonic temperature. Please add this variable to the list. Fig. 2, 3 etc.: Please mention in the MM part which software you used to create these graphs. Line 498: Replace "ground conduction heat"

with "ground heat flux" Line 239: Typo: "if" not "It" Line 257-258: Please rewrite this sentence. This sentence is unreadable. Line 323: From here on the numbering of the figures is wrong. In this line, for example, your refer to Fig. 8 not to Fig. 9.

---

## Author Comment (AC2) · 10 Mar 2017

We thank the Reviewer for the positive revision of this manuscript, and for contributing to its improvement. According to their general comments, we added a discussion on the impact the exclusion of the soil heat storage term has on the surface energy balance. We tried to answer every comment in detail.

General comments The authors evaluated a 15-year EC data record of a savanna FLUXNET site in Kruger National Park. This is a great and unique dataset. The authors focus in their analysis on the surface energy balance closure and energy partitioning.

The topic fits very well into the scope of HESS, and the dataset will be interesting for a broad readership of HESS. The dataset is carefully evaluated for several aspect. The authors give interesting insight to technical problems that showed up over the 15 years, and they analyzed, among others, the effect of the season as well as the friction velocity on the EBR. My major concern is related to the measurement of the ground heat flux. Firstly, important information is missing. As far as I got it, the authors did not determine the heat storage change in the layer above the heat flux plate (HFP). Please state this clearly in the Material and Method (MM) part. Response: Thank you for your comment. The authors have included this information. Line 169-171: "We did not account for the heat storage terms in the EBR, including soil and canopy heat storage, and energy storage by photosynthesis and respiration, in this study. The significance of neglecting these storage terms will be discussed."

Moreover, it remains unclear how the three HFP readings were averaged. Two were installed under tree canopies and one at open space. How did you compute the mean ground heat flux representative for the footprint? Did you compute a weighted mean or did you compute simply the mean over the three plates. In the latter case the ground heat flux would be systematically underestimated, because the areal fraction of tree canopy is only 30%. Response: Thank you for your comment. The soil heat flux for the site was computed as a weighted mean of the three measurements, and this information has been included: Line 149-151: "Soil heat flux was then computed as a weighted mean of the three measurements, i.e., two taken under tree canopies and one on open space."

Secondly, in general neglecting the heat storage term must result to a certain extent in a systematic underestimation of the ground heat flux and hence to a systematic overestimation of the available energy and consequently to an underestimation of the EBR. From own measurements (HFP were installed in 8 cm depth) I know that this storage term can reach at unshaded surfaces 50 to 100 W/m2. Please discuss in detail the magnitude of error that might originate from your methodological approach.

Response: Thank you for your comment. In this study, the authors did not consider the heat storage terms, and have included this information in the methodology (see above response). Furthermore, we have included a discussion on the expected error that could result for this omission as suggested by the Reviewer. Line 304-317: While G plays a significant role on the surface energy balance closure, our study ignored the different energy storage terms in determining the EBR, including the soil heat storage term. The exclusion of this storage term results in the underestimation of G, as the real value of G is a combination of the flux measured by the plate and the heat exchange between the ground and the depth of the plate. This in turn contributes to overestimating the available energy, which then lowers the EBC. As reported by different studies, the omission of the soil heat storage results in the underestimation of the energy EBC by up to 7 %. For instance, Zuo et al. (2011) reported an improvement of 6 to 7 % when they included the soil heat storage in their calculation of EBR, at the Semi-Arid Climate and Environment Observatory of Lan-Zhou University (SACOL) site in semi-arid grassland over the Loess Plateau of China. In their study in the three sites in the Badan Jaran desert, Li, Liu, Wang, Miao, and Chen (2014) analysed the effect of including soil heat storage derived by different methods in the energy balance closure; their EBR improved by between 1.5 % and 4 %. The improvement of the EBR in the study in a FLUXNET boreal site in Finland by Sánchez, Caselles, and Rubio (2010) was shown to be 3 % when the soil heat storage was included, which increased to 6 % when other storage terms (canopy air) were taken into account.

Moreover, I wondered why the authors do not give any information on monthly or annual evapotranspiration rates (in mm). With that information one could get a guess of the climatic water balance of that ecosystem. I think this would be very interesting for the reader and would further strengthen the manuscript. Response: Thank you for the comment. This manuscript focuses on the surface energy balance and how solar radiation is partitioned in this savanna site. The evapotranspiration part has been covered in a different manuscript.
Specific comments Line 132: Please state which software tool (e.g. TK3 or EddyPro) was used to process the EC raw data. Response: Thank you for your comment. The information was added: Line 129-130: "The Eddysoft software was used to process the raw data collected from the eddy covariance system (Kolle & Rebmann, 2007)."

Line 135: How did you detect outliers? Please explain! Response: Thank you for your comment. The information was added: Line 144-145: "The data outliers were detected using the outlier detection procedure found in the Statistica software."

Line 170: The intention of Figure 1 is to show temperature, VPD and rainfall anomalies between the years. I think this way of displaying the data is not really optimal for this purpose. The authors should think about a better way to present these anomalies. One way could be to compute for every month the difference from the 15- year mean and list these differences in a table (rows: month; column: years). Months, for example, that were warmer than the 15-year average get a red color, months that were colder get a blue color. The larger the difference the more intensive the red and blue color. Response: Thank you for your comment. Figure 1 has been redone as shown below.

Line 198: This is a little bit data cosmetic. The very good EBR is achieved thanks to the really bad year 2013, which had an EBR of 3.76. If you remove this year as outlier the mean EBR reduces to 0.77. I suggest that the authors start this chapter with explaining the technical problems that showed up over the years with the very low EBR and the extremely high EBR in 2013. And after that the authors should refer only to the years with no data or technical issues. Response: Thank you for your comment. The authors first reported the yearly EBR, as well as the mean multiyear EBR of all years, including those with low quality data. Here we also explained the technical problems that resulted in the low EBR. Thereafter, we stated that further analysis excluded the years with low quality data (Line 232-233).

Figure 2: In the OLS approach, the dependent variable (turbulent fluxes) is plotted against the independently derived available energy. See for example Wilson et al.

[Figure]

(2002). If you plot it the other way round, as you did, the slope of the regression does not fit to the EBR. If the EBR is below one than the slope must also be below one. In the year 2007, for example, the EBR is 0.44 but the slope is 1.46. That does not fit together. Moreover, if you use the turbulent flux in a regression as independent variable your statistical model assumes that this variable has no error. Please correct everywhere the figures and update the numbers for slopes and intercepts! Response: Thank you for your comment. The authors have rectified this on Figures 2 to 5.

Line 205: Here it would be important for the reader to know how you modeled the incoming and outgoing longwave radiation, so that they can avoid this mistake in future. Please describe this model in more detail. Response: Thank you for your comment. The authors do not have the detailed information on the methodology used to model incoming and outgoing longwave radiation during the 2004-2008 period. However, different methods that model net radiation from climatic variables (Irmak, Mutiibwa, & Payero, 2011; Ortega-Farias, Antonioletti, & Olioso, 2000; Sabziparvar & Mirgaloybayat, 2015) and remote sensing based methods (Kjaersgaard et al., 2009; Samani, Bawazir, Bleiweiss, Skaggs, & Tran, 2007; Sun et al., 2013; Wu et al., 2017) have been developed. It would also be of interest to evaluate these models using the Skukuza eddy covariance data, as an extension to this study.

Technical comments Line 28: Avoid the wording bad and good data. Please use instead e.g. low- and high quality data Response: Corrected, thank you.

Line 40: I would not count energy stored in ground as a minor flux term (see above). Please rephrase. Response: The sentence has been rephrased as: Line 42: "...heat stored by the canopy, the ground and energy storage terms by photosynthesis."

Line 83: If you start the sentence with first I expect that there comes a second item. Response: The succeeding sentence was started as follows: Line 87: "Then, we examined how the surface energy partitioning...."

Line 150: Replace "incorrect assumption" with "simplification". Response: Corrected,

Line 163.

Line 148: Introduce here the symbol "R2". Response: Done (Line 161), thank you.

Line 158: Rewrite "4" in "four". Response: Done (Line 173), thank you.

Line 224: Here it is unclear which storage term was included by Sanchez et al. (2010). Please rewrite! Response: Thank you for your comment. This reference has been moved to the section which explains the effect of including storage terms in the EBR.

Table 1: Why clayey? In the MM part you write that the texture ranged from sand to loamy sand. Please check! Response: Thank you for your comment. The soil type has been removed in Table 1 to avoid confusion.

Table 1: Campbell Scientific is not the manufacturer of the HFP. The manufacturer is Huskeflux. Please correct that and mention whether you used self-calibrating plates or not. Response: We used the HFT3 soil heat flux plate, which was manufactured by Campbell Scientific, not the HFP soil heat flux plate, a product of Huskeflux.

Table 1: Beside the wind speed the anemometer measures also the sonic temperature. Please add this variable to the list. Response: Added, thank you.

Fig. 2, 3 etc.: Please mention in the MM part which software you used to create these graphs. Response: Mentioned (Line 173-174), thank you.

Line 498: Replace "ground conduction heat" with "ground heat flux" Response: Corrected (Line 543), thank you.

Line 239: Typo: "if" not "It" Response: Edited (Line 262), thank you.

Line 257-258: Please rewrite this sentence. This sentence is unreadable. Response: Edited, thank you. Line 282-283: To understand the effect of friction velocity on the energy balance closure, surface energy data which had corresponding friction velocity (u*) data, were analysed.

[Figure]

Line 323: From here on the numbering of the figures is wrong. In this line, for example, you refer to Fig. 8 not to Fig. 9. Response: Corrected, thank you.

References Irmak, S., Mutiibwa, D., & Payero, J. O. (2011). Net radiation dynamics: Performance of 20 daily net radiation models as related to model structure and intricacy in two climates. Transactions of the ASABE, 53(4), 1059-1076.

Kjaersgaard, J. H., Cuenca, R. H., Martínez-Cob, A., Gavilán, P., Plauborg, F., Mollerup, M., & Hansen, S. (2009). Comparison of the performance of net radiation calculation models. Theoretical and Applied Climatology, 98(1), 57-66. doi:10.1007/s00704-008-0091-8

Kolle, O., & Rebmann, C. (2007). EddySoft: Dokumentation of a Software Package to Acquire and Process Eddy Covariance Data.

Li, Y., Liu, S., Wang, S., Miao, Y., & Chen, B. (2014). Comparative study on methods for computing soil heat storage and energy balance in arid and semi-arid areas. Journal of Meteorological Research, 28, 308-322.

Ortega-Farias, S., Antonioletti, R., & Olioso, A. (2000). Net radiation model evaluation at an hourly time step for Mediterranean conditions. Agronomie, 20(2), 157-164.

Sabziparvar, A., & Mirgaloybayat, R. (2015). Evaluation of Some Existing Empirical and Semi-Empirical Net Radiation Models for Estimation of Daily ET0. Journal of Advanced Agricultural Technologies Vol, 2(1).

Samani, Z., Bawazir, A. S., Bleiweiss, M., Skaggs, R., & Tran, V. D. (2007). Estimating daily net radiation over vegetation canopy through remote sensing and climatic data. Journal of Irrigation and Drainage Engineering, 133(4), 291-297.

Sánchez, J. M., Caselles, V., & Rubio, E. M. (2010). Analysis of the energy balance closure over a FLUXNET boreal forest in Finland. Hydrology and Earth System Sciences, 14(8), 1487-1497.

Sun, Z., Gebremichael, M., Wang, Q., Wang, J., Sammis, T. W., & Nickless, A. (2013). Evaluation of clear-sky incoming radiation estimating equations typically used in remote sensing evapotranspiration algorithms. Remote Sensing, 5(10), 4735-4752.

Wu, B., Liu, S., Zhu, W., Yan, N., Xing, Q., & Tan, S. (2017). An Improved Approach for Estimating Daily Net Radiation over the Heihe River Basin. Sensors, 17(1), 86.

Zuo, J.-q., Wang, J.-m., Huang, J.-p., Li, W., Wang, G., & Ren, H. (2011). Estimation of ground heat flux and its impact on the surface energy budget for a semi-arid grassland. Sci Cold Arid Region, 3, 41-50.

Please also note the supplement to this comment:
http://www.hydrol-earth-syst-sci-discuss.net/hess-2016-685/hess-2016-685-AC2-supplement.pdf

[revised manuscript text omitted]

---

## Author Response (AR2)

[revised manuscript text omitted]

**Reviewer #1**

**General**

The article presents an impressive multi-year dataset of energy fluxes over an under sampled part of the world. The focus is on the energy balance closure.

> *We thank the reviewer for his thorough and positive evaluation of this manuscript; his positive feedback has contributed to its improvement. Further analysis of G was done to investigate how it impacts the surface energy balance closure, as recommended by the reviewer. We hope that this effort will improve the manuscript, by strengthening the weak points highlighted by the Reviewer. We tried to respond to the comments of each reviewer with as much detail as possible to the best of our ability.*

**Major remarks**

The article points out the difficulties of collecting valid data over long periods of time.

My first question is if the cleaning procedures may have introduced any biases? For example, periods with rainfall often produce problems with sonic anemometers. Could you comment on this?

> **Response:** *Thank you for your comment.*
>
> *When measuring the different variables using the eddy covariance system, apart from instrument failure, instruments like the sonic anemometer and the net radiometer are affected by different phenomena, like rainfall events and wind gusts, resulting in faulty diagnostic signals, outliers and data gaps, which are sources of error and bias. Thus data cleaning, which involves screening, diagnosing and editing, of these half-hourly surface energy data, was done to reduce bias and error.*
>
> *In our study we used the Amelia II software, an R-program designed to impute missing data using Expectation-Maximization with Bootstrapping (EMB) multiple imputation algorithm (Honaker et al., 2011). This program resamples the original dataset using bootstrapping, where it then imputes the missing data The iterations done in this algorithm ensure that any bias is limited, if not completely eliminated.*

The second is a pet peeve of mine and concerns the ground heat flux. If I understand correctly (the text is not so clear, see also below under minor remarks), you report EBR per half hour. Over the period of half an hour, ground heat flux can typically play an important role. Ground heat flux is also not very well captured with ground heat flux plates, which basically measure the temperature difference between the top and bottom of a piece of plastic in the ground. Even if the plates would work as intended, they are clearly biased as 2/3 of the plates are under canopies while only 30% of the area has a canopy. I don't ask for you to go back and repeat the measurements with better measurements of G but a critical discussion is needed. A simple way to get some idea is to compare half hourly results with daily averaged EBRs. G will generally be negligible at daily scales while it can easily make up 50% of the energy balance at a half hourly basis.

> *Response: Thank you for the comment.*
>
> *The authors agree that soil heat flux plays a significant role on the surface energy balance, as it determines the amount of energy available for the turbulent fluxes. In this study, however, we did not do detailed investigation of the influence G has on the surface energy balance, as this would be a subject of*

*study on its own, especially in this study area. We, hence, only highlighted the effect G has on the surface*
*energy balance by calculating how the exclusion of G in the EBR computation ((H+LE)/Rn) affects the*
*results compared with the initial EBR ((H+LE)/(Rn-G)) values. The results reported as follows:*
*Line 300-305: Soil heat flux (G) plays a significant role in the surface energy balance as it determined*
*how much energy is available for the turbulent fluxes, especially in areas with limited vegetation cover.*
*In this study, we examined how G, i.e., its presence or absence, impacts on the EBR. Our results revealed*
*a decrease of up to 7 %, with an annual mean of 3.13±2.70, in EBR when G was not included in the*
*calculation. During the daytime, the absence of G resulted in a decrease of approximately 10 % of the*
*initial EBR, while at night-time EBR was as low as 50 % of the initial EBR, showing that G has greater*
*impact on the surface energy balance at night.*
*Also, the G used was a weighted mean of the three measurements to avoid any biases associated with*
*the fact that 2/3 of the plates are under canopies while only 30% of the area is on bare ground.*
Finally, the article would become ten times more valuable if you make the (cleaned?) dataset available online.
***Response:*** *Noted, thank you.*
*The issue of publishing this dataset will be discussed with all parties involved.*
**Minor remarks**
Line 29: Winter & summer are not so obvious terms for people not familiar with Kruger National Park. Either use
months or, my preference, talk about dry and wet, as you do later under 2.1.
***Response:*** *Summer changed to wet, and winter changed to dry (Line 29, 30).*
Line 36: characterized by or rather correlated with?
***Response:*** *Thank you, this has been changed (Line 37).*
Line 41: Is the heat stored in the ground not the ground heat flux G?
***Response:*** *Thank you for your comment.*
*The heat stored referred to in this context is the heat exchange between the ground and the depth of the*
*plate, and not the flux measured by the soil heat flux plate.*
Line 47: Potential evapotranspiration is a problematic term. Better use "reference evaporation".
***Response:*** *Changed, thank you (Line 48).*
Line 58: "measured" instead of "measurable"
**Response:** Changed, thank you (Line 59).
Line 92: Here you use Earth, elsewhere earth. I have no preference but best stick to one.
***Response:*** *Noted, thank you.*
Line 117: canopies instead of canopiesa

*Response: Corrected, thank you (Line 122).*

Line 125: Did you use any software? Is code available?

*Response: The Eddysoft software was used to process the raw data (Line 129).*

Line 126: You state that all upward fluxes are positive but later you clearly change this in Equation 1 and also when you state that daytime Rn is positive.

*Response: The statement has been removed.*

Line 157: I surmised that you evaluated the dataset by looking at half hourly EBRs. The text here is, however, not very clear on that. Please make explicit.

*Response: The sentence now reads:*

*Line 173: "...the half-hourly data were separated..."*

Line 198: Is the 0.11 the standard deviation in the estimate of the mean? Or is it the standard deviation? Also, with EBR always being larger than zero, perfect at one, and not upwardly bounded, would a logarithmic averaging scheme not make more sense?

*Response: Thank you for your comment.*

*±0.11 is the standard deviation.*

*Our results show a few of the EBR values above 1, i.e. 2010-2012, December-February and September-November, and the 25 and 100 percentiles, and the rest of the values are below zero. This is in line with other studies that show that EBR is almost always less than 1, i.e. the measured available energy is larger than the sum of the measured turbulent fluxes, as shown by different studies (Chen et al., 2009; Were, Villagarcía, Domingo, Alados-Arboledas, & Puigdefábregas, 2007; Wilson et al., 2002; Xin & Liu, 2010; Yuling, 2005). These studies also alluded to the concern within the micrometeorological community that the turbulent fluxes (LE + H) are frequently (though not always) underestimated by about 10–30% relative to estimates of available energy (Rn-G), making the EBR less than one.*

Lines 213 and further and in general throughout this part: You mix literature review with results. It is more common not to introduce too much additional information from outside the study past the introduction. Would probably be better to move this to intro (but don't make it too long!).

*Response: Thank you for your observation.*

*The authors agree that literature is mixed with the results. The results section is combined with the discussion, hence the literature citations are found in this section.*

Line 230 and further: The Results and discussion focus on EBR and other outcomes in a very descriptive way. Would be better to already include more physical insights here as to why you see what you see.

*Response: Thank you for your observation.*

*The descriptive way shown here is the explanation of the results, since the Results and Discussion sections are combined.*

Line 264: Why the hurry? Here also please expand on role of G as mentioned above.

***Response:*** *Thank you for the comment.*

*The authors have included how G, its inclusion and non-inclusion, impacts on the value of EBR. This*

*was fully explained above.*

Lines 334 and further: In general, there is a bit of a mix between the focus on EBR and the more general and the probably more interesting general interpretation of results. The article is built up around EBR and only towards the end do general energy & water availability considerations come up. Perhaps point to these earlier in the text.

In any case, please shift the perspective from starting with other studies, such as by Gu et al., and comparing those with your results to a perspective that starts with your results and then compares those, preferably a bit more systematically, with other studies.

***Response:*** *Thank you for the observation.*

*The authors would like to point out that this study focuses on two issues, i.e. the energy balance closure*

*first, then how the available energy is partitioned over time in this ecosystem, based on the climate*

*conditions in the region, particularly, precipitation (a proxy of soil water availability), VPD and Rn*

*impact on this partitioning.*

**References**

Chen, S., Chen, J., Lin, G., Zhang, W., Miao, H., Wei, L., . . . Han, X. (2009). Energy balance and partition in inner mongolia steppe ecosystems with different land use types. *Agricultural and Forest Meteorology,*

*149*(11), 1800-1809.

Were, A., Villagarcía, L., Domingo, F., Alados-Arboledas, L., & Puigdefábregas, J. (2007). Analysis of effective resistance calculation methods and their effect on modelling evapotranspiration in two different patches of vegetation in semi-arid se spain. *Hydrology and Earth System Sciences Discussions, 11*(5), 1529-

1542.

Wilson, K., Goldstein, A., Falge, E., Aubinet, M., Baldocchi, D., Berbigier, P., . . . Field, C. (2002). Energy balance closure at fluxnet sites. *Agricultural and Forest Meteorology, 113*(1), 223-243.

Xin, X., & Liu, Q. (2010). The two-layer surface energy balance parameterization scheme (tsebps) for estimation of land surface heat fluxes. *Hydrology and Earth System Sciences, 14*(3), 491-504.

Yuling, F. U. (2005). Energy balance closure at chinaflux sites.

 **Reviewer #2**

We thank the Reviewer for the positive revision of this manuscript, and for contributing to its improvement.

According to their general comments, we added a discussion on the impact the exclusion of the soil heat storage term has on the surface energy balance. We tried to answer every comment in detail.

**General comments**

The authors evaluated a 15-year EC data record of a savanna FLUXNET site in Kruger National Park. This is a great and unique dataset. The authors focus in their analysis on the surface energy balance closure and energy partitioning. The topic fits very well into the scope of HESS, and the dataset will be interesting for a broad readership of HESS. The dataset is carefully evaluated for several aspect. The authors give interesting insight to technical problems that showed up over the 15 years, and they analyzed, among others, the effect of the season as well as the friction velocity on the EBR.

My major concern is related to the measurement of the ground heat flux.

Firstly, important information is missing. As far as I got it, the authors did not determine the heat storage change in the layer above the heat flux plate (HFP). Please state this clearly in the Material and Method (MM) part.

*Response: Thank you for your comment.*

*The authors have included this information.*

*Line 169-171: "We did not account for the heat storage terms in the EBR, including soil and canopy*

*heat storage, and energy storage by photosynthesis and respiration, in this study. The significance of*

*neglecting these storage terms will be discussed."*

Moreover, it remains unclear how the three HFP readings were averaged. Two were installed under tree canopies and one at open space. How did you compute the mean ground heat flux representative for the footprint? Did you compute a weighted mean or did you compute simply the mean over the three plates. In the latter case the ground heat flux would be systematically underestimated, because the areal fraction of tree canopy is only 30%.

*Response: Thank you for your comment.*

*The soil heat flux for the site was computed as a weighted mean of the three measurements, and this*

*information has been included:*

*Line 149-151: "Soil heat flux was then computed as a weighted mean of the three measurements, i.e.,*

*two taken under tree canopies and one on open space."*

Secondly, in general neglecting the heat storage term must result to a certain extent in a systematic underestimation of the ground heat flux and hence to a systematic overestimation of the available energy and consequently to an underestimation of the EBR. From own measurements (HFP were installed in 8 cm depth) I know that this storage term can reach at unshaded surfaces 50 to 100 W/m2. Please discuss in detail the magnitude of error that might originate from your methodological approach.

*Response: Thank you for your comment.*

*In this study, the authors did not consider the heat storage terms, and have included this information in*

*the methodology (see above response). Furthermore, we have included a discussion on the expected*

*error that could result for this omission as suggested by the Reviewer.*

*Line 304-317: While G plays a significant role on the surface energy balance closure, our study ignored the different energy storage terms in determining the EBR, including the soil heat storage term. The exclusion of this storage term results in the underestimation of G, as the real value of G is a combination of the flux measured by the plate and the heat exchange between the ground and the depth of the plate. This in turn contributes to overestimating the available energy, which then lowers the EBC. As reported by different studies, the omission of the soil heat storage results in the underestimation of the energy EBC by up to 7 %. For instance, Zuo et al. (2011) reported an improvement of 6 to 7 % when they included the soil heat storage in their calculation of EBR, at the Semi-Arid Climate and Environment Observatory of Lan-Zhou University (SACOL) site in semi-arid grassland over the Loess Plateau of China. In their study in the three sites in the Badan Jaran desert, Li, Liu, Wang, Miao, and Chen (2014) analysed the effect of including soil heat storage derived by different methods in the energy balance closure; their EBR improved by between 1.5 % and 4 %. The improvement of the EBR in the study in a FLUXNET boreal site in Finland by Sánchez, Caselles, and Rubio (2010) was shown to be 3 % when the soil heat storage was included, which increased to 6 % when other storage terms (canopy air) were taken into account.*

Moreover, I wondered why the authors do not give any information on monthly or annual evapotranspiration rates (in mm). With that information one could get a guess of the climatic water balance of that ecosystem. I think this would be very interesting for the reader and would further strengthen the manuscript.

> **Response***: Thank you for the comment.*
>
> *This manuscript focuses on the surface energy balance and how solar radiation is partitioned in this savanna site. The evapotranspiration part has been covered in a different manuscript.*

**Specific comments**

Line 132: Please state which software tool (e.g. TK3 or EddyPro) was used to process the EC raw data.

> **Response:** *Thank you for your comment. The information was added:*
>
> *Line 129-130: "The Eddysoft software was used to process the raw data collected from the eddy covariance system (Kolle & Rebmann, 2007)."*

Line 135: How did you detect outliers? Please explain!

> **Response:** *Thank you for your comment. The information was added:*
>
> *Line 144-145: "The data outliers were detected using the outlier detection procedure found in the Statistica software."*

Line 170: The intention of Figure 1 is to show temperature, VPD and rainfall anomalies between the years. I think this way of displaying the data is not really optimal for this purpose. The authors should think about a better way to present these anomalies. One way could be to compute for every month the difference from the 15- year mean and list these differences in a table (rows: month; column: years). Months, for example, that were warmer than the 15-year average get a red color, months that were colder get a blue color. The larger the difference the more intensive the red and blue color.

***Response:*** *Thank you for your comment.*

*Figure 1 has been redone as shown below.*

[Figure]

**Figure 11: Summary of mean monthly anomalies (a) air temperature, (b) VPD, and (c) rainfall from 2000**

**to 2014**

Line 198: This is a little bit data cosmetic. The very good EBR is achieved thanks to the really bad year 2013, which had an EBR of 3.76. If you remove this year as outlier the mean EBR reduces to 0.77. I suggest that the authors start this chapter with explaining the technical problems that showed up over the years with the very low EBR and the extremely high EBR in 2013. And after that the authors should refer only to the years with no data or technical issues.

**Response:** Thank you for your comment.

> *The authors first reported the yearly EBR, as well as the mean multiyear EBR of all years, including those with low quality data. Here we also explained the technical problems that resulted in the low EBR. Thereafter, we stated that further analysis excluded the years with low quality data (Line 232-233).*

Figure 2: In the OLS approach, the dependent variable (turbulent fluxes) is plotted against the independently derived available energy. See for example Wilson et al. (2002). If you plot it the other way round, as you did, the slope of the regression does not fit to the EBR. If the EBR is below one than the slope must also be below one. In the year 2007, for example, the EBR is 0.44 but the slope is 1.46. That does not fit together. Moreover, if you use the turbulent flux in a regression as independent variable your statistical model assumes that this variable has no error. Please correct everywhere the figures and update the numbers for slopes and intercepts!

> *Response: Thank you for your comment.*
>
> *The authors have rectified this on Figures 2 to 5.*

[Figure]

**Figure 11̶2̶: 15-year series of annual regression analysis of turbulent (sensible and latent) heat fluxes against**

**available energy (net radiation minus ground heat flux) from 2000 to 2014 at Skukuza, (SA). The colour**

**bars represent the count of EBR values.**

[Figure]

Figure 123: Seasonal turbulent fluxes (H+LE) correlation to available energy (Rn-G) for Skukuza flux tower from summer (Dec-Feb), autumn (March-May), winter (June-Aug), spring (Sept-Nov). The colour bars represent the count of EBR values

[Figure]

Figure 134: Turbulent fluxes correlation to available energy for daytime (a) and night-time (b), using the full (2000-2014) 15-year available data series. The colour bars represent the count of EBR values

[Figure]

**Figure 145: OLS and EBR evaluations at different friction velocity sorted at four quartiles. The colour bar represents the count of EBR values. The colour bars represent the count of EBR values.**

Line 205: Here it would be important for the reader to know how you modeled the incoming and outgoing longwave radiation, so that they can avoid this mistake in future. Please describe this model in more detail.

    *Response: Thank you for your comment.*

    *The authors do not have the detailed information on the methodology used to model incoming and outgoing longwave radiation during the 2004-2008 period. However, different methods that model net radiation from climatic variables (Irmak, Mutiibwa, & Payero, 2011; Ortega-Farias, Antonioletti, & Olioso, 2000; Sabziparvar & Mirgaloybayat, 2015) and remote sensing based methods (Kjaersgaard et al., 2009; Samani, Bawazir, Bleiweiss, Skaggs, & Tran, 2007; Sun et al., 2013; Wu et al., 2017) have been developed. It would also be of interest to evaluate these models using the Skukuza eddy covariance data, as an extension to this study.*

**Technical comments**

Line 28: Avoid the wording bad and good data. Please use instead e.g. low- and high quality data

*Response: Corrected, thank you.*

Line 40: I would not count energy stored in ground as a minor flux term (see above). Please rephrase.

*Response: The sentence has been rephrased as:*

*Line 42: "...heat stored by the canopy, the ground and energy storage terms by photosynthesis."*

Line 83: If you start the sentence with first I expect that there comes a second item.

*Response: The succeeding sentence was started as follows:*

*Line 87: "Then, we examined how the surface energy partitioning...."*

Line 150: Replace "incorrect assumption" with "simplification".

*Response: Corrected, Line 163.*

Line 148: Introduce here the symbol "R2".

*Response: Done (Line 161), thank you.*

Line 158: Rewrite "4" in "four".

*Response: Done (Line 173), thank you.*

Line 224: Here it is unclear which storage term was included by Sanchez et al. (2010). Please rewrite!

*Response: Thank you for your comment.*

*This reference has been moved to the section which explains the effect of including storage terms in the*

*EBR.*

Table 1: Why clayey? In the MM part you write that the texture ranged from sand to loamy sand. Please check!

*Response: Thank you for your comment.*

*The soil type has been removed in Table 1 to avoid confusion.*

Table 1: Campbell Scientific is not the manufacturer of the HFP. The manufacturer is Huskeflux. Please correct that and mention whether you used self-calibrating plates or not.

*Response: We used the HFT3 soil heat flux plate, which was manufactured by Campbell Scientific, not*

*the HFP soil heat flux plate, a product of Huskeflux.*

Table 1: Beside the wind speed the anemometer measures also the sonic temperature. Please add this variable to the list.

*Response: Added, thank you.*

*Fig. 2, 3 etc.: Please mention in the MM part which software you used to create these graphs.*

*Response: Mentioned, thank you.*

Line 498: Replace "ground conduction heat" with "ground heat flux"

*Response: Corrected, thank you.*

Line 239: Typo: "if" not "It"

*Response: Edited, thank you.*

Line 257-258: Please rewrite this sentence. This sentence is unreadable.

*Response: Edited, thank you.*

*Line 282-283: To understand the effect of friction velocity on the energy balance closure, surface energy*

*data which had corresponding friction velocity (u\*) data, were analysed.*

Line 323: From here on the numbering of the figures is wrong. In this line, for example, you refer to Fig. 8 not to

Fig. 9.

*Response: Corrected, thank you.*

***Response:*** *Thank for your comment.*

*The authors have revised the manuscript according to the responses given. The changes are highlighted*

*on the edited manuscript.*

    *The authors concur that although soil heat flux (G) is the smallest component of the surface*

*energy balance (SEB), it plays a significant role in the SEB as it determines the amount of energy*

*available for the turbulent fluxes. In some instances, G has been ignored based on the assumption that*

*G is generally negligible at daily scales on the basis that the heat stored during the day is released during*

*the night. Neglecting G in the surface energy balance, however, results in the overestimation of the*

*available energy and higher energy balance ratio (EBR). A handful of studies have focused on G,*

*including its measurement techniques, how it varies with soil temperature and moisture, and the role it*

*plays on the surface energy balance (Heusinkveld et al. 2004; Russell et al. 2015; Santanello Jr & Friedl,*

*2003; Sauer & Horton, 2005; Yang & Wang, 2008).*

    *Heat storage terms, which include soil and canopy heat storage, and energy storage by*

*photosynthesis and respiration were not accounted for in our study. Neglecting the soil heat storage term*

*has a significant effect on the surface energy balance, as the real value of G is a combination of the flux*

*measured by the plate and the heat exchange between the ground and the depth of the plate. This results*

*in the underestimation of G and hence to a systematic overestimation of the available energy and*

*consequently to an underestimation of the EBR. Soil heat storage term varies with the depth of the soil*

*heat flux plate, as demonstrated by Ochsner et al. (2006), who reported that at a depth of 1 cm, the*

*maximum G is up to 13% less than the maximum surface value, and at 10 cm maximum G is up to 70%*

*less than the surface value, so an exclusion of this term results in high error margin. Meyers and Tilden*

*(2004) showed that the ground heat storage term was as high as 40 W/m² early in the day, which is a*

*considerable amount to be ignored, and Heusinkveld et al. (2004) proved that failure to account for the*

*storage term can cause errors of 10–200 W/m² for bare soils or under sparse vegetation. Liu et al. (2017)*

*reported an increase in OLS slope of an average 8.8% and a mean daily EBR increase of 5% when the*

*soil heat storage term was considered in their study.*

*Hence, a study that investigates the role of G and the different heat storage terms in the surface*

*energy balance using this dataset would contribute to further understanding this component of the SEB*

*in semi-arid savanna areas.*

*With this in mind, to address the editor and reviewers comments on the importance of G and*

*uncertainties related with the exclusion of the soil heat storage term, the authors have:*

i.  *Highlighted the impact G has on the surface energy balance closure, as depicted by the energy*

*balance ratio (EBR), by showing how the inclusion ((H+LE)/(Rn-G)) and exclusion of G*

*((H+LE)/Rn) varies the EBR values. This was done using the entire dataset, as well as on the day-*

*night-time datasets.*

ii.  *Reviewed in detail the magnitude of error that might originate from the exclusion of the soil heat*

*storage term in our study area based on different studies.*

*The above have been added to the manuscript:*

*Line299-330: Soil heat flux (G) plays a significant role in the surface energy balance as it determined*

*how much energy is available for the turbulent fluxes, especially in areas with limited vegetation cover.*

*Its exclusion in the surface energy balance results not only in the overestimation of the available energy,*

*but also the overestimation of the EBR. Its exclusion in surface energy balance studies results not only*

*in the overestimation of the available energy, but also the overestimation of the EBR. In this study, we*

*examined how inclusion and exclusion of G impacts the surface energy balance closure. When G was*

*excluded in the calculation, the multiyear EBR ranged between 0.73 and 1.07, with an annual mean EBR*

*of 0.90±0.11, which is about 3 % lower than the initial EBR (0.93±0.11). While the initial daytime EBR*

*was 0.96, it was 0.87 when G was excluded, which is a decrease of approximately 10 %. The nighttime*

*EBR was 0.13, as low as 50 % of the initial EBR (0.26), showing that G has greater significance on the*

*surface energy balance at night. These results are in agreement with other studies, for instance, Ogee et*

*al., (2001) showed that soil heat flux represents up to 50% of net radiation at midday and up to 80%*

*during night-time. Stull (2012) also reported that during daytime G only accounts for 5-15% of net*

*radiation, whereas at night, it is up to 50%.*

*While G is an important component of the SEB, our study ignored the different energy storage*

*terms in determining the EBR, including the soil heat storage term. The exclusion of the soil heat storage*

*term results in the underestimation of G, as the real value of G is a combination of the flux measured by*

*the plate and the heat exchange between the ground and the depth of the plate. This in turn contributes*

*to the overestimation of the available energy, which then lowers the EBC. Among other factors*

*(vegetation cover, soil moisture and temperature), this storage term varies with the depth of the soil heat*

*flux plate as demonstrated by Ochsner et al. (2006), who reported that at a depth of 1 cm, the maximum*

*G is up to 13% less than the maximum surface value, and at 10 cm maximum G is up to 70% less than*

*the surface value, thus its exclusion results in similar error margins in the EBC. As reported by different*

*studies, the omission of the soil heat storage results in the underestimation of the energy EBC by up to*

*7%. For instance, Liu et al. (2017) reported an increase in OLS slope of an average 8.8% and a mean*

*daily EBR increase of 5% when the soil heat storage term was considered in their study in the Taihu*

*Lake region of the Southern China Plain. In their study in the three sites in the Badan Jaran desert, Li*

*et al. (2014) analysed the effect of including soil heat storage derived by different methods in the energy*

*balance closure; their EBR improved by between 1.5 % and 4 %. Zuo et al. (2011) reported an*

*improvement of 6 to 7 % when they included the soil heat storage in their calculation of EBR, at the*

*Semi-Arid Climate and Environment Observatory of Lan-Zhou University (SACOL) site in semi-arid*

*grassland over the Loess Plateau of China. The improvement of the EBR in the study in a FLUXNET*

*boreal site in Finland by Sánchez et al. (2010) was shown to be 3 % when the soil heat storage was*

*included, which increased to 6 % when other storage terms (canopy air) were taken into account.*

2) A clear answer is expected if the dataset will/ is published or not. What is the strategy?

 *Response: Thank for your comment.*

 *The dataset will be published with the manuscript. However, if it is used for any research purpose this*

 *publication must be cited.*

3) I also expect that future responses to the reviewer comments are made in a more structured way. Normally, answers to the reviewer comments are highlighted. This is not the case with your responses, which makes the document hard to read.

 *Response: Thank for your comment.*

 *The authors have tried to address the comments in a more structured way.*